# Structure of the complete *Saccharomyces cerevisiae* Rpd3S-nucleosome complex

Jonathan W. Markert[1], Seychelle M. Vos [2] ✉ & Lucas Farnung [1] ✉

Acetylation of histones is a key post-translational modification that guides gene expression regulation. In yeast, the class I histone deacetylase containing Rpd3S complex plays a critical role in the suppression of spurious transcription by removing histone acetylation from actively transcribed genes. The *S. cerevisiae* Rpd3S complex has five subunits (Rpd3, Sin3, Rco1, Eaf3, and Ume1) but its subunit stoichiometry and how the complex engages nucleosomes to achieve substrate specificity remains elusive. Here we report the cryo-EM structure of the complete Rpd3S complex bound to a nucleosome. Sin3 and two copies of subunits Rco1 and Eaf3 encircle the deacetylase subunit Rpd3 and coordinate the positioning of Ume1. The Rpd3S complex binds both trimethylated H3 tails at position lysine 36 and makes multiple additional contacts with the nucleosomal DNA and the H2A–H2B acidic patch. Direct regulation via the Sin3 subunit coordinates binding of the acetylated histone substrate to achieve substrate specificity.

The small histone deacetylase reduced potassium dependency 3 (Rpd3S) complex plays a critical role in the regulation of gene expression in the yeast *S. cerevisiae* and is involved both in the activation and repression of genes in a variety of cellular processes including metabolism, meiosis, heat shock, and osmotic stress[1–3]. Rpd3S is part of the SIN3-HDAC family of histone deacetylase complexes that are conserved from yeast to humans[4] and is a five-subunit complex composed of the class I histone deacetylase (HDAC) subunit Rpd3 and the additional subunits Eaf3, Rco1, Sin3, and Ume1[5,6]. As part of the Set2-Rpd3S pathway in *S. cerevisiae*, Set2 dependent histone H3 lysine 36 (H3K36) methylation directs the Rpd3S complex to coding genomic regions[7,8]. Interaction with RNA polymerase II and transcription elongation factors fine-tune Rpd3S occupancy on actively transcribed genes[9]. Rpd3S deacetylates histones H3 and H4 to prevent cryptic transcription initiation events from within gene bodies because an accumulation of histone acetylation can lead to the destabilization of transcribed nucleosomes, ultimately resulting in the loss of nucleosomes from the gene body[5,10–16].

Rpd3S is directed to actively transcribed genes by binding of the Rpd3S subunit Eaf3 to nucleosomes via an association of the Eaf3 chromodomain (CHD) with methylated H3K36[6,12]. The specific recruitment to H3K36 methylated nucleosomes is additionally coupled to another Rpd3S subunit, Rco1[17]. Recently, structures of the *H. sapiens* SIN3B and the *S. cerevisiae* Rpd3L complex have been determined[18,19]. These structures have provided the first structural snapshots of the architecture of SIN3-HDAC complexes. How SIN3-HDAC complexes engage their nucleosomal substrate to achieve a broad substrate specificity for acetylated H3 and H4 histones, however, remains elusive.

Here we provide the cryogenic electron microscopy (cryo-EM) structure of the complete Rpd3S complex bound to a nucleosome at an overall resolution of 2.8 Å. The structure reveals the architecture of the Rpd3S complex with a core formed by subunits Rpd3, Sin3, Rco1, and Eaf3, and an auxiliary module that is composed of Ume1 and second copies of Rco1 and Eaf3. Rpd3S recognizes the nucleosome via multiple histone and DNA contacts. The Rco1 subunit binds the H2A–H2B acidic patch and the core histone fold of H3. Both copies of Eaf3 bind the trimethylated H3K36. Additionally, Rpd3S positions itself on the nucleosomal substrate by interacting with nucleosomal DNA at super helical locations (SHLs) 1, 1.5, 2, 6.5 and extranucleosomal DNA at SHL 7.5 and 8. Binding of Sin3 into the catalytic tunnel of Rpd3 and modulation of Sin3 binding by Rco1 provides additional levels of regulatory control to achieve substrate specificity in a chromatin environment. Our structure explains why the Rpd3S complex does not

[1]Department of Cell Biology, Blavatnik Institute, Harvard Medical School, Boston, MA, USA. [2]Department of Biology, Massachusetts Institute of Technology, Cambridge, MA, USA. ✉e-mail: seyvos@mit.edu; lucas_farnung@hms.harvard.edu

require inositol phosphate for enzymatic activity as Sin3 binds an important surface of Rpd3 that is occupied by inositol phosphate in other class I deacetylases. Interestingly, the Ume1 subunit associates with the Eaf3 subunit of the Rpd3S complex in a manner that is distinct from its association with the related Rpd3L complex. Together, our structure elucidates how Rpd3S engages a mononucleosome and provides the structural basis for understanding how the Ume1 subunit is coordinated within Rpd3S.

## Results

### Rpd3S-nucleosome complex assembly and cryo-EM

To obtain the structure of the nucleosome bound Rpd3S complex, we recombinantly expressed and purified *S. cerevisiae* Rpd3S with subunits Sin3, Rpd3, Ume1, Rco1, and Eaf3 (Fig. 1a) in insect cells using a

baculovirus expression system (Supplementary Fig. 1a). We generated a nucleosomal substrate with a modified Widom 601 sequence and 30 and 31 base pairs of extranucleosomal DNA on the DNA entry and exit side (Methods, Supplementary Fig. 1b-d). The nucleosome contains a chemically generated analogue of histone H3 with lysine residue 36 trimethylation (H3K36Cme3)[20], and histone H4 proteins with lysine residue 16 acetylation[21] (Methods).

We formed a stable complex between Rpd3S and the nucleosome, purified the complex by size-exclusion chromatography, and selected fractions containing the assembled Rpd3S-nucleosome complex (Supplementary Fig. 1e, f, Methods). The selected fractions underwent mild cross-linking using glutaraldehyde, followed by dialysis, and subsequent cryo-EM preparation. The sample was then subjected to single particle cryo-EM analysis on a Titan Krios microscope. We were

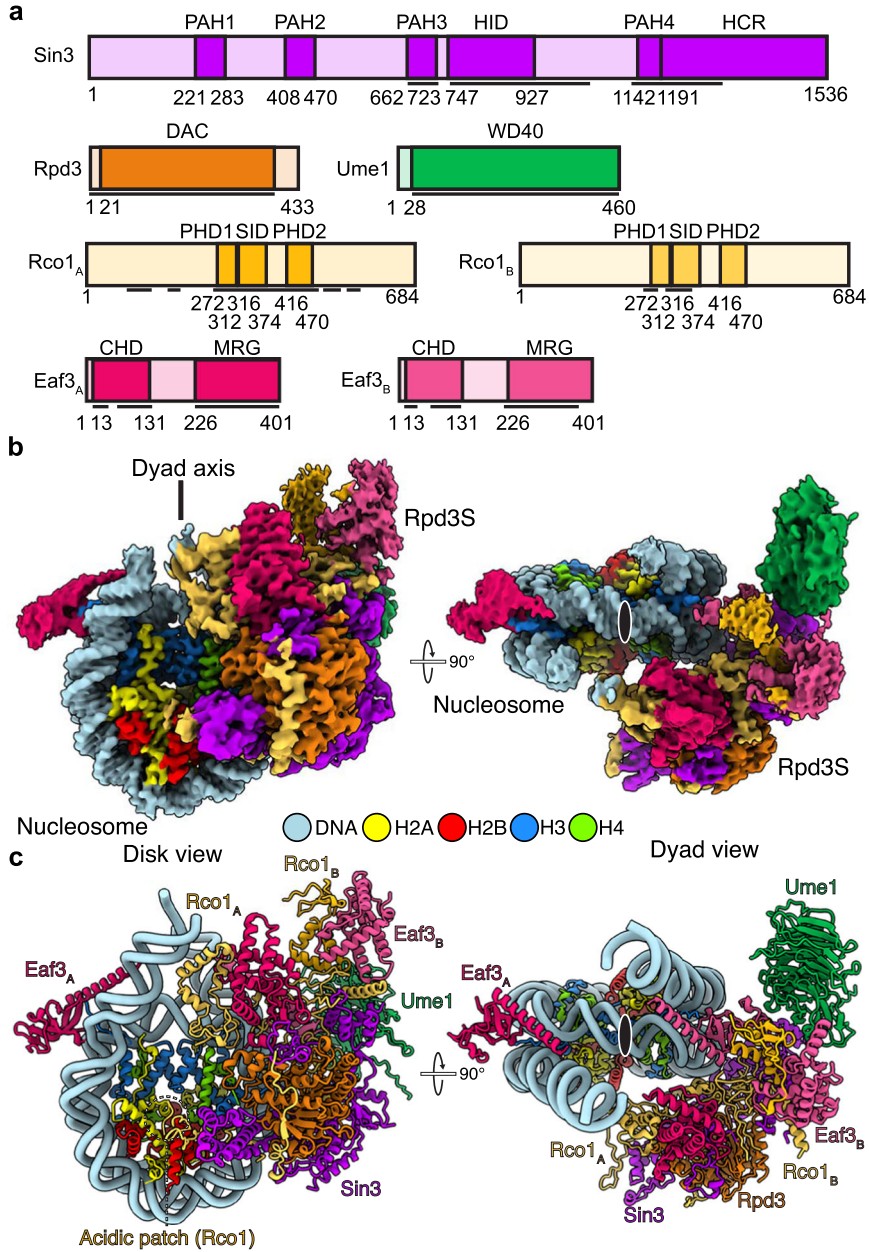

**Fig. 1 | Structural overview of the nucleosome bound Rpd3S complex. a** Domain architecture of Rpd3S subunits. Domain borders are indicated. Black lines indicate modelled regions. Domain boundaries are indicated by residue number. **b** Composite single-particle reconstruction of Rpd3S-nucleosome complex (map H). DNA, H2A, H2B, H3, H4, Sin3, Rpd3, Ume1, Rco1₍A₎, Eaf3₍A₎, Rco1₍B₎, and Eaf3₍B₎ are colored in light blue, yellow, red, dodger blue, chartreuse, purple, orange, green, dark yellow, dark red, light yellow, and light red, respectively. Color code used throughout. Ume1 density is shown from map G. **c** Atomic model of Rpd3S-nucleosome complex. Dyad axis indicated with black oval on white background.

able to reconstruct the Rpd3S-nucleosome complex from 481,141 particles, with an overall resolution of 2.8 Å (Fig. 1, Supplementary Figs. 2–4, Supplementary Table 1, Supplementary Movie 1). Additional masked refinements yielded reconstructions of the nucleosome and the Rpd3S complex at resolutions of 2.7 Å and 2.9 Å, respectively (Supplementary Fig. 2). We docked a known crystal structure of the nucleosome into the density and manually built the extranucleosomal DNA[22]. AlphaFold2 predictions for the Rpd3S subunits were fit into corresponding densities. The model was subsequently inspected, manually rebuilt, and real-space refined (Supplementary Fig. 4, Methods). Together, this resulted in an atomic model of the complete Rpd3S-nucleosome complex with good stereochemistry (Fig. 1c, Supplementary Tables 2, 3).

Our model suggests that the Rpd3S complex is comprised of single copies of Sin3, Rpd3, and Ume1, alongside two copies of Rco1 and Eaf3 (Fig. 1). The Rpd3S complex containing two copies of Rco1 is consistent with in vitro data[17]. It was previously unknown that two copies of Eaf3 were present within the Rpd3S complex. This observation, however, is consistent with previous findings that the Rpd3S complex preferentially engages dinucleosomal substrates[16]. In our structure, we observe binding of the Eaf3 chromodomains to the H3K36Cme3 analog of both H3 histones on both sides of the nucleosome (Fig. 1b, c).

## Overall architecture of the complete Rpd3S complex

The Rpd3S complex is positioned above one face of the nucleosomal disk (Fig. 2). The Rpd3S core module consists of Rpd3, Sin3, one copy of Rco1 and Eaf3 (Fig. 2a). Our cryo-EM map enables the confident modeling of side-chain residues for these components (Supplementary Table 3). The core module positions the Rpd3 histone deacetylase subunit centrally and positions it above the N-terminal tail of H4 (Fig. 2a). Rpd3 adopts a fold that is highly similar to other class I HDAC enzymes consisting of a single α/β domain that contains an eight-stranded parallel β-sheet sandwiched between 13 α-helices[23]. Importantly, the catalytic tunnel of Rpd3 with the deacetylase active site is oriented towards the nucleosomal disk. All other subunits of Rpd3S are arranged around Rpd3 and form an extensive network of inter-subunit contacts (Fig. 2a).

Sin3 scaffolds the Rpd3S complex[17] and envelopes Rpd3 with its paired amphipathic helix domain 3 (PAH3), its middle domain, the HDAC interacting domain (HID), and the C-terminal region of Sin3 that encompasses a region (Sin3 residues 966-1142) N-terminal of the PAH4 domain, the PAH4 domain itself and the highly conserved region (HCR) (Figs. 1a, 2b). The arrangement of Rpd3 in relation to Sin3 resembles the interaction observed in the Rpd3L complex[18] (Supplementary Fig. 5). In contrast to other class I HDACs, Rpd3 does not interact with inositol phosphate[19,24]. Instead Sin3 HID residues Glu811 and Glu812 insert into the Rpd3 basic patch and form an electrostatic network with Rpd3 residues Lys41, Arg280, and Arg316 (Fig. 2c, Supplementary Fig. 6). Consequently, the presence of Sin3 makes the requirement of inositol phosphate obsolete in the Rpd3S complex. Like the *H. sapiens* SIN3B complex, a loop ("gate loop") in the Sin3 HID inserts into the catalytic tunnel of Rpd3[19] (Fig. 2d, e).

Beyond Sin3, one copy of Rco1 (Rco1$_A$) serves as an additional scaffold and plays a key role in organizing the Rpd3S core module (Fig. 3). The N-terminus of Rco1$_A$ (Rco1$_A$ residues 78-130) interacts with the Sin3 HID domain (Fig. 3b, Supplementary Fig. 5), mirroring the interaction seen between Sds3 and the Sin3 HID domain in the Rpd3L complex[18,25]. Rco1$_A$ meanders across the surface of Rpd3 via residues 163-190 and complements Rpd3's β-sheet by contributing an additional β-strand (Rco1$_A$ residues 176-179) (Fig. 3c). Notably, in the Rpd3L complex, Dep1 engages in similar contacts across the Rpd3 surface but does not augment the Rpd3 β-sheet[18,25] (Supplementary Fig. 5). Via its PHD finger 1 (PHD1) and Sin3 interacting domain (SID), Rco1$_A$ forms a heterodimer with one copy of Eaf3 (Eaf3$_A$) (Fig. 3d). This interaction is

mediated by the Eaf3 MORF4-related gene (MRG) domain and rationalizes how deletion of the Rco1$_A$ SID results in dissociation of Eaf3 from the Rpd3S complex[26]. A Rco1$_A$ region connecting the SID domain and PHD2 contact the Sin3 gating loop. The Rco1$_A$ PHD2 packs against Rpd3, the Sin3 C-terminal region, and Sin3 PAH3 domain (Fig. 3e). The C-terminus of Rco1$_A$ then forms the interface with the second copy of Rco1 (Rco1$_B$) and Eaf3 (Eaf3$_B$) (Fig. 3f). The additional copies of Rco1 and Eaf3 form the auxiliary module that sits near the extranucleosomal DNA on the Sin3 side of the nucleosome (Fig. 2a).

The interface of Rco1$_B$ and Eaf3$_B$ is less resolved, so AlphaFold2 predictions of these domains were rigid body docked into the secondary features visualized in our cryo-EM map (Supplementary Table 3). This interface appears highly similar to the Rco1$_A$ and Eaf3$_A$ interface and is again formed by the Eaf3 MRG domain and the Rco1 PHD1 and SID (Fig. 3d, f).

## Ume1 is positioned between Sin3 and the Eaf3$_B$ chromodomain

During cryo-EM analysis, we discovered an additional density between the Eaf3$_B$ chromodomain and the Sin3 HCR (Supplementary Figs. 2, 7). Further classification and refinements revealed density that resembled a WD40 repeat domain-like fold (Supplementary Fig. 2). Using the *S. pombe* Sin3S complex as an initial model[27], we identified this extra density as the Ume1 subunit. While our resolution is currently insufficient for accurate Ume1 positioning, we can clarify its global placement relative to other Rpd3S subunits (Supplementary Table 3). Recent structures of Rpd3L and *S. pombe* Sin3S show that Ume1 is primarily coordinated by the Sin3 HCR domain[18,25]. The Ume1-binding part of the Sin3 HCR domain is only flexibly tethered to the Rpd3S core, rationalizing why we can observe Ume1 only at limited resolutions of greater than 8 Å. In addition to Ume1 being in close proximity to the Sin3 HCR domain, we observe that Ume1 is positioned near the chromodomain of Eaf3$_B$. Specifically, the backside of the Eaf3$_B$ chromodomain is oriented towards the WD40 repeat domain of Ume1 (Supplementary Fig. 7). Our data explain previous genetic and biochemical observations where it was observed that the Sin3 PAH3 domain was essential for Ume1 association with Rpd3S[28]. The PAH3 domain coordinates Rco1 and Eaf3$_B$ of the auxiliary module[16,17], suggesting that the Ume1 positioning we observe near Eaf3$_B$ could contribute to Ume1 incorporation into Rpd3S[28] in the context of the nucleosome but is not necessarily required[29]. Ume1, Rpd3, and Sin3 can form a complex in the absence of Eaf3[29].

## Rpd3S-nucleosome contacts

The distinct architecture of the Rpd3S complex facilitates various interactions between its constituent subunits and both nucleosomal and extra-nucleosomal DNA and histones (Fig. 4a). These interactions are mediated via subunits Sin3, Rco1, and Eaf3.

Firstly, Sin3 contacts the DNA phosphate backbone at SHL 2 via the C-terminal region of Sin3 (Fig. 4b). Specifically, Sin3 residues Lys940 and Lys1244 and Gln937 and Gln1222 contact both strands of the nucleosomal DNA via electrostatic interactions at SHL 2.

Secondly, Rco1$_A$ contacts extranucleosomal DNA via its SID (residues Lys320, Lys321, and Lys 328) at SHL 8 (Fig. 4c). This interaction of Rco1$_A$ rationalizes the requirement for the Rco1 SID and extranucleosomal DNA for nucleosomal substrate binding by the complex[7,17]. Due to the asymmetric arrangement of the auxiliary Rco1$_B$/Eaf3$_B$ module, the interaction is not mirrored by the Rco1$_B$ SID. Additionally, we observe Rco1$_A$ arginine 384 is positioned near the phosphate backbone of the nucleosomal DNA at SHL -1.5.

Thirdly, Eaf3 interfaces with both nucleosomal DNA and extranucleosomal DNA (Fig. 4d, e). Specifically, both Eaf3 copies contact nucleosomal DNA at SHL + 6.5 and −6.5 via lysine residue 85 in the Eaf3 chromodomains (Fig. 4d, e). The Eaf3 chromodomain lysine residue 26 contacts the nucleosomal DNA at SHL 1.5. Furthermore, a helix in the Eaf3 chromodomains projects towards the linker DNA on both sides of

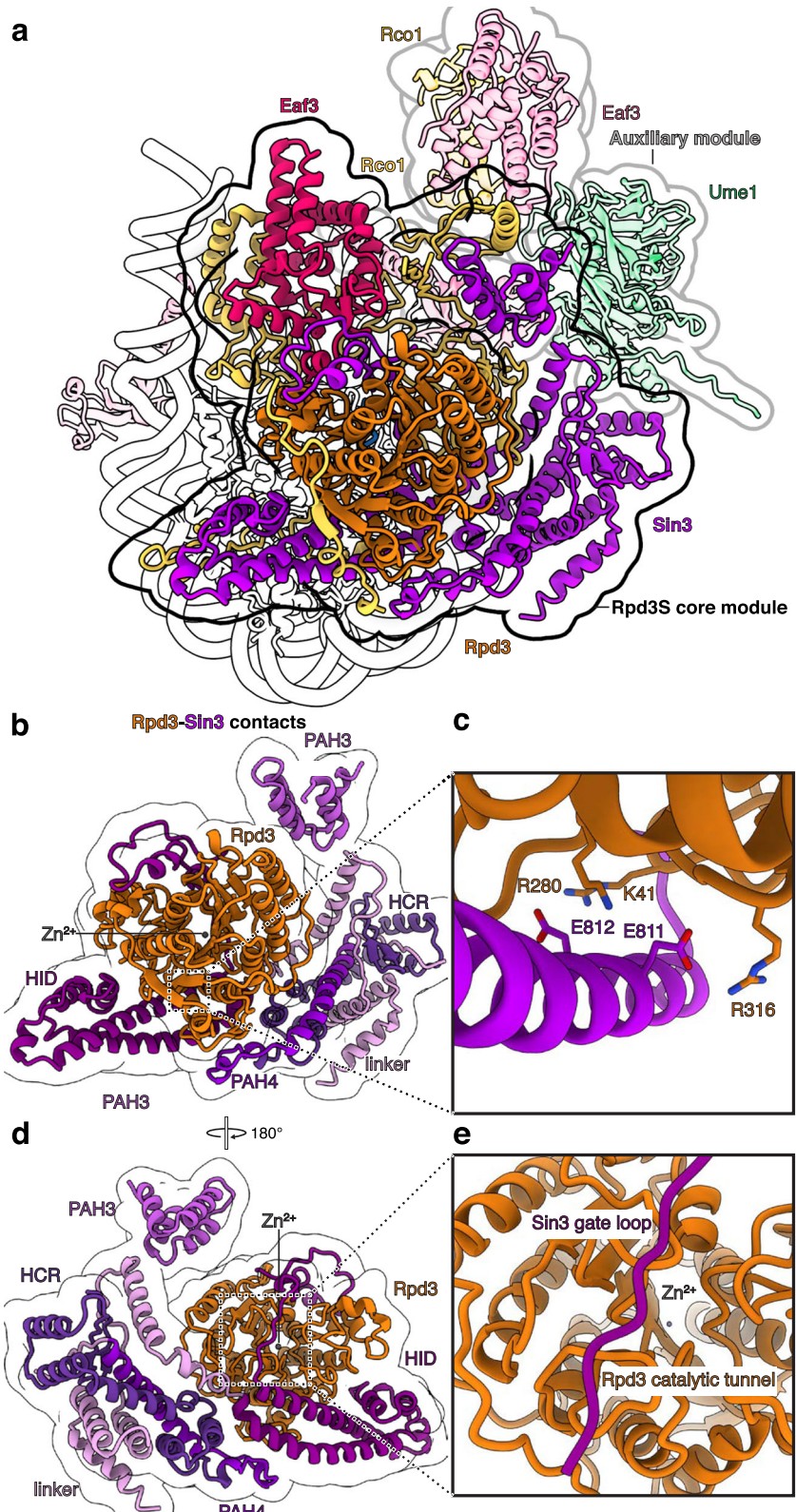

**Fig. 2 | Rpd3S modules and Sin3-Rpd3 interactions. a** Overview highlighting the Rpd3S core module with Rpd3, Sin3, Rco1$_A$, and Eaf3$_A$, and the auxiliary module with Rco1$_B$, Eaf3$_B$, and Ume1. **b** Sin3 encircles Rpd3 and interacts with the HDAC fold of Rpd3 via multiple domains. **c** Detailed interaction of Sin3 via residues Sin3 residues Glu811 and Glu812 with Rpd3 residues Lys41, Arg280, and Arg316 reveals allosteric activation of Rpd3 by Sin3. **d** View as in b with 180° rotation. **e** The Sin3 gating loop binds the Rpd3 HDAC fold.

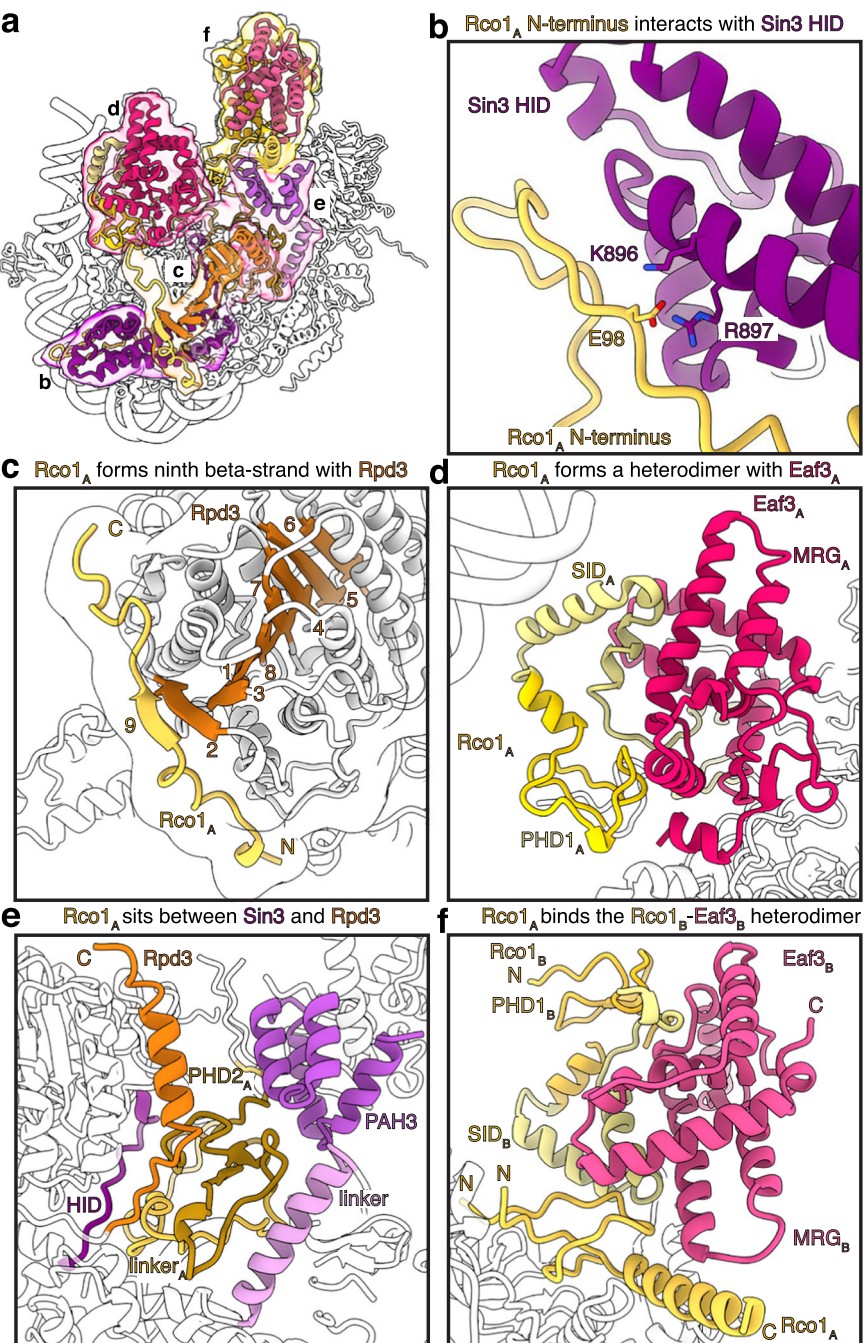

**Fig. 3 | Rco1 interactions in the Rpd3S complex. a** Overview of Rco1 interactions with Sin3, Rpd3, and Eaf3. Interfaces shown in b-f are highlighted. **b** Interaction of the Rco1$_A$ N-terminus with the Sin3 HID. Interacting residues Rco1$_A$ Glu98 and Sin3 Lys896 and Arg897 are shown as sticks. **c** Rco1$_A$ complements the eight-strand β-sheet of Rpd3. **d** Rco1$_A$ SID and PHD1 form a heterodimer with Eaf3$_A$. **e** Rco1$_A$ PHD2 and the linker region of Rco1$_A$ are located below Rpd3 and interact with the Sin3 linker, gating loop (HID), and PAH3. **f** The Rco1$_B$–Eaf3$_B$ interface is similar to the Rco1$_A$–Eaf3$_A$ heterodimerization interface. Additionally, Rco1$_A$ packs against the Rco1$_B$ SID and the MRG domain of Eaf3$_B$.

the nucleosome with lysine residues 129 and 130 placed near the extranucleosomal DNA at SHL ± 7.5 (Fig. 4d, e, Supplementary Fig. 8a). The low resolution within this region limits confidence that a direct contact with DNA occurs via Rco1$_A$ lysine residues 129 and 130, but their relative placement rationalizes previous observations that Rpd3S requires extranucleosomal DNA for optimal binding[7]. This interaction induces a ~30° bend in the extra-nucleosomal DNA (Supplementary Fig. 8b, c).

Rpd3S also interacts with the H2A–H2B acidic patch. We identified residues interacting with the H2A–H2B acidic patch on the Rpd3S-facing side of the nucleosomal disc (Fig. 4f). The cryo-EM density was,

however, ambiguous in this region and could only be broadly assigned to the N-terminus of Rco1. To identify the interface, we used Alpha-Fold2 multimer predictions of the N-terminal Rco1 with H2A–H2B and identified a Rco1 region encompassing residues 30-65 as a likely candidate for H2A–H2B acidic patch binding (Supplementary Fig. 5d, e). The binding is mediated by a canonical arginine anchor that inserts into the acidic patch[30] and forms interactions with H2A residues Glu61, Asp90, and Glu92 (Fig. 4f). It is possible that the acidic patch serves as an anchor for Rpd3S to both promote recruitment to the nucleosome and enhance deacetylase activity, a model that is consistent with other known histone deacetylase-nucleosome interactions[31,32].

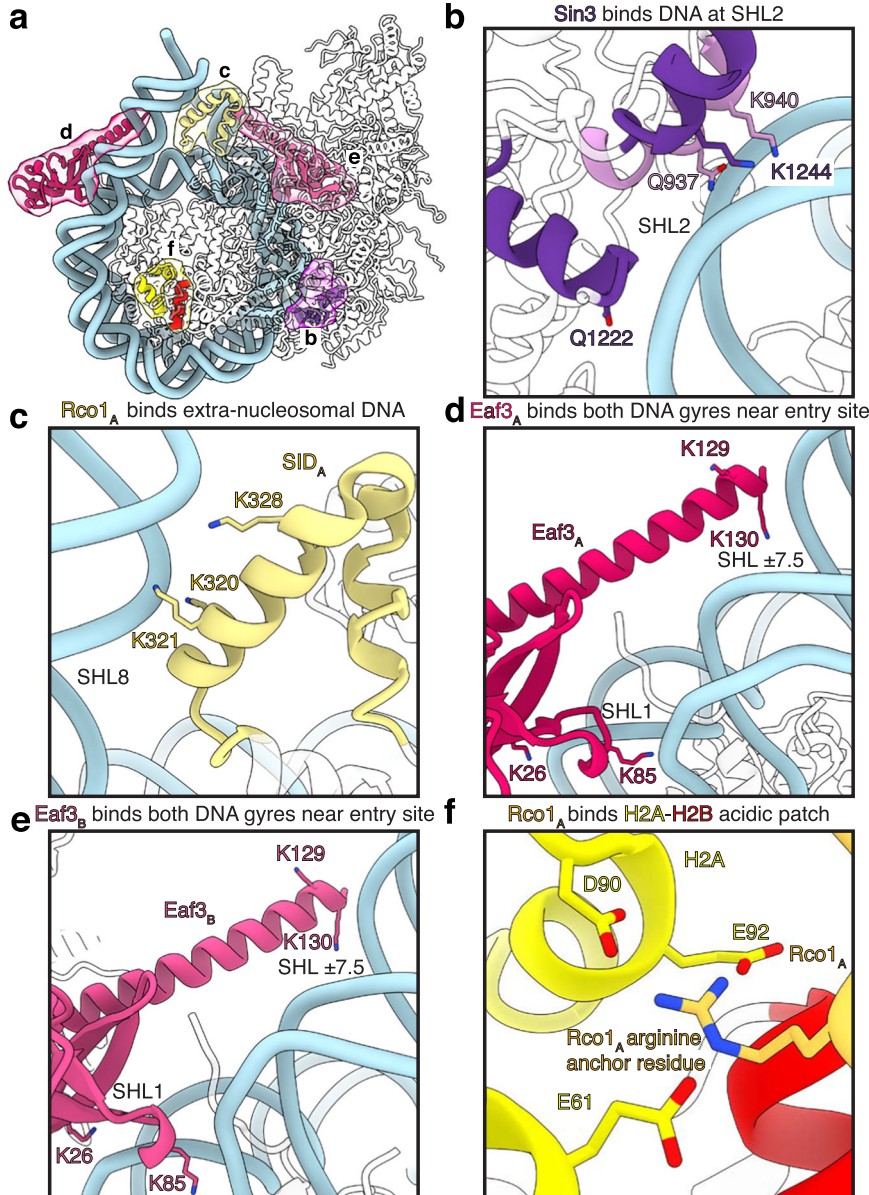

**Fig. 4 | The Rpd3S complex forms extensive contacts with the nucleosome.**
**a** Overview of Sin3, Rco1, and Eaf3 interaction interfaces with nucleosomal, extra-nucleosomal DNA, H3 and the H2A–H2B acidic patch. Interfaces shown in b-f are highlighted. **b** Sin3 contacts the phosphate backbone of nucleosomal DNA at SHL 2. **c** Rco1$_A$ interacts with the phosphate backbone of extranucleosomal DNA at SHL 8.
**d** Eaf3$_A$ contacts both DNA gyres at SHL 1 and 7.5 and is positioned near extra-nucleosomal DNA at SHL8. **e** Eaf3$_B$ interacts with nucleosomal DNA similar to EAF3$_A$. **f** Rco1 interacts via an N-terminal arginine residue with the H2A-H2B acidic patch. Interaction of the arginine residue with H2A residues Glu61, Asp90, and Glu92 are shown as sticks. Heteroatoms are colored canonically.

Together, these interactions position the Rpd3S complex above the N-terminal tail of histone H4 and in close proximity to the H3 N-terminal tail, explaining how the Rpd3S complex can achieve a wide substrate specificity and deacetylate a broad range of acetylated residues on both H3 and H4.

### Eaf3 interaction with methylated H3K36

The recruitment of the Rpd3S complex to regions of active gene transcription is facilitated by the interaction between the Eaf3 chromodomain and the trimethylated lysine residue 36 of histone H3 (Fig. 5a, b, Supplementary Fig. 9a). Our structure highlights the binding of both Eaf3 copies to the trimethylated histone tails. Situated at SHL ± 1, the Eaf3 chromodomains interact with nucleosomal DNA at SHL ± 1 and on the opposite DNA gyre at SHL ± 6.5. This interaction across DNA gyres is reminiscent of the engagement of the LEDGF PWWP domain with the trimethylated H3K36 nucleosome[33]

(Supplementary Fig. 9b). Interaction with both DNA gyres positions the Eaf3 chromodomains proximal of the H3 tail, allowing the tri-methylated H3K36C analogue to insert into the Eaf3 chromodomain aromatic cage that is formed by Eaf3 residues His18, Tyr23, Tyr81, Trp84, and Trp88 (Fig. 5a, b). The binding of nucleosomal DNA and the trimethylated histone tail is identical for both Eaf3 chromodomains (Supplementary Fig. 9a). Interestingly, both Eaf3 chromodomains interact with the same nucleosome in our structure. It is plausible, however, that one of the Eaf3 chromodomains can engage neighboring H3K36me3 nucleosomes. This mechanism could explain how the Rpd3S complex can execute processive deacetylation of nucleosomes by spreading to neighboring nucleosomes[16].

### Interaction of the Rpd3S active site with an H3/H4 substrate

Our cryo-EM analysis revealed density for a seven amino acid long peptide in the active site of Rpd3 (Fig. 5c). The density is of sufficiently

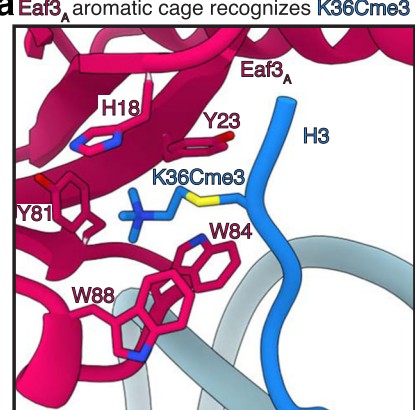

**a** Eaf3$_A$ aromatic cage recognizes K36Cme3

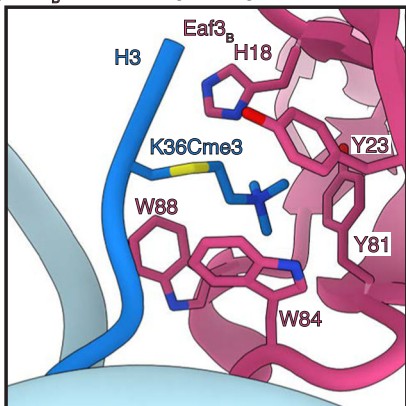

**b** Eaf3$_B$ aromatic cage recognizes K36Cme3

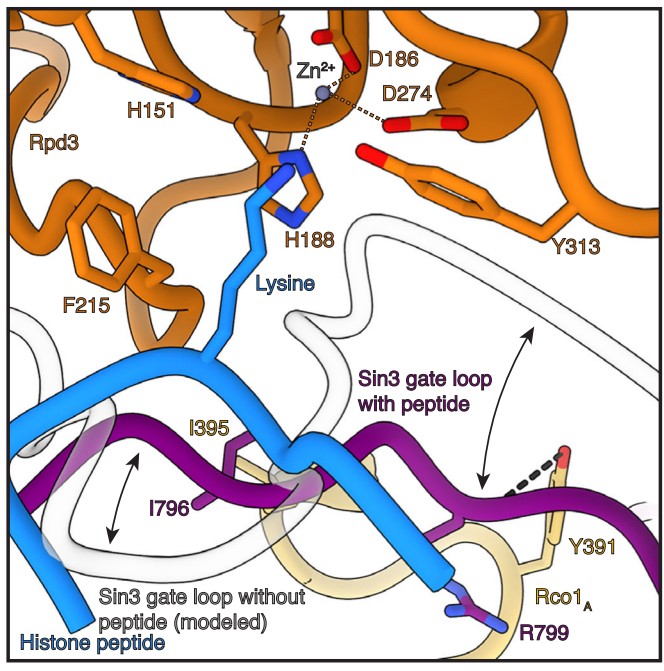

**c** Sin3 gate loop flipping allows histone lysine to insert into Rpd3 active site

**Fig. 5 | The Eaf3 chromodomains bind the H3K36Cme3 analogue. a, b** The chromodomain of Eaf3$_A$ and Eaf3$_B$ bind the H3K36Cme3 analogue via an aromatic cage formed by Eaf3 residues His18, Tyr23, Tyr81, Trp84, and Trp88. The binding mode for both H3K36Cme3 analogues to Eaf3 is conserved. **c** The peptide lysine residue inserts into the active site. Upon binding, the Sin3 gating loop undergoes a conformational rearrangement to accommodate the substrate peptide. Rco1$_A$ further stabilizes the rearranged gating loop. Important interactions are indicated. Zn$^{2+}$ ion in the Rpd3 active site and its coordination is indicated.

high quality that we can assign a centrally located lysine residue that projects towards the Rpd3 active site and three amino acids of unknown identity on the N- and C-terminal site of the lysine residue. This heterogeneity is expected as Rpd3S has a broad substrate specificity for lysine residues of the H3 and H4 tails, resulting in a heterogeneous mixture of possible H3 and H4 substrates bound by the Rpd3 active site. Consistent with the binding of the peptide, the Sin3 gate loop is in a flipped position that facilitates binding of the peptide[19] (Fig. 5c). Like the *H. sapiens* SIN3B complex, the gate loop participates in stabilizing the peptide substrate in the Rpd3 active site[19]. Importantly, the Sin3 gate loop is additionally stabilized by hydrogen bonds between the backbone of Rco1$_A$ Tyr391 and Sin3 Arg799 (Fig. 5c). Additionally, residues connecting the SID and PHD2 of Rco1$_A$ form a hydrophobic pocket that the Sin3 gating loop residue Ile796 inserts into (Fig. 5c). Conversely, the Sin3 gate loop is likely positioned to fully cover the catalytic tunnel of Rpd3 and prevent association of any substrate within the active site of Rpd3 in the absence of a histone substrate[19].

## Discussion

Here we report the structure of the complete Rpd3S complex engaging a nucleosomal substrate. Our structure describes the subunit composition of the Rpd3S complex with one copy of Rpd3, Sin3, and Ume1, and two copies of Rco1 and Eaf3. While we do not have any evidence to suggest additional copies of subunits present in the Rpd3S complex, future work will focus on fully characterizing the Rpd3S stoichiometry in higher-order chromatin contexts. Our findings demonstrate the importance of Sin3 and Rco1 as constitutive scaffolding proteins in the Rpd3S complex. We show how DNA and histone contacts between Rpd3S and the nucleosome position Rpd3 above the nucleosome to guarantee broad substrate specificity. Additionally, the structural work clarifies allosteric mechanisms of Sin3 binding to Rpd3 to drive binding of appropriate substrates and achieve substrate specificity. Together, our structure provides an overview of a complete Sin3-HDAC complex with all its subunits bound to its nucleosomal substrate.

Our structure clarifies the core architecture of Sin3-HDAC complexes bound to a nucleosomal substrate. The centrally located

histone deacetylase subunit is positioned above the nucleosomal disk, Sin3/SIN3 is organized around the deacetylase, and Rco1 (or structural homologues) bridge towards additional subunits that can then provide nucleosome specificity. In contrast to other HDACs such as the MiDAC[34], SMRT HDAC3[35] or NuRD[36] complexes, Sin3-containing complexes do not require the co-factor inositol phosphate as Sin3 substitutes for inositol phosphate[19,36]. Binding of Sin3 in the catalytic tunnel of Rpd3 provides additional regulatory control over the binding of histone substrates. As *H. sapiens* Sin3-HDAC complexes are attractive targets of novel drug inhibitors against human diseases[37], the Rpd3S complex presents an initial framework that can guide novel small-molecule inhibitor studies. Going forward, this initial work will help to elucidate how the Rpd3S complex acts co-transcriptionally in a chromatin environment.

During manuscript preparation, multiple preprints and one peer-reviewed manuscript reporting Rpd3S-nucleosome complex structures missing the Ume1 subunit became available[24,38,39]. Overall, these structures corroborate our findings and support the significance of extensive multivalent nucleosome interactions in facilitating effective deacetylation. Notably, Guan et al. describe the Rpd3S core as identical to ours and demonstrates the importance of Rco1$_A$ in deacetylating the H3 and H4 N-terminal tails, but is missing the Rco1$_A$-acidic patch interaction and Ume1[39].

## Methods
No statistical methods were used to predetermine sample size. The experiments were not randomized, and the investigators were not blinded to allocation during experiments and outcome assessment.

### Cloning and expression of the Rpd3S complex
*S. cerevisiae* Sin3, Rpd3, Ume1, Rco1, and Eaf3 were amplified from *S. cerevisiae* genomic DNA from strain BJ5464 and cloned into the MacroLab 438 vector series for expression in insect cells using ligation-independent cloning. The MacroLab vector series is based on pFastBac vectors. Rpd3 was cloned into the 438-C vector with an N-terminal His6 tag, followed by a maltose binding protein (MBP) tag, a 10 amino acid linker, and a TEV protease cleavage site. All other subunits were cloned into 438-A vectors with no tag. All subunits were then combined into the pBigBac1a vector[40]. Bacmid, virus, and protein production were then performed as previously described[41]. The Rpd3S complex was expressed in Hi5 insect cells.

### Purification of the Rpd3S complex
The Rpd3s complex was purified from 600 mL Hi5 insect cells. Cells were resuspended in lysis buffer (300 mM NaCl, 20 mM Na·HEPES pH 7.4 at 25 °C, 10% (v/v) glycerol, 30 mM imidazole pH 8, 1 mM TCEP pH 8, 0.284 µg ml$^{-1}$ leupeptin, 1.37 µg ml$^{-1}$ pepstatin A, 0.17 mg ml$^{-1}$ PMSF and 0.33 mg ml$^{-1}$ benzamidine) and lysed by sonication. Lysed cells were subsequently centrifuged and then ultra-centrifuged. The supernatant was then cleared through a 0.45 µM filter, and then loaded onto a 5 mL HisTrap HP (Cytiva) column equilibrated in lysis buffer. After 10 CV washes with lysis buffer, a self-packed XK column (Cytiva) with 15 mL of Amylose resin (New England Biolabs) was attached to the HisTrap column. Sample was directly eluted onto the Amylose column with 25 mL nickel elution buffer (300 mM NaCl, 20 mM Na·HEPES pH 7.4 at 25 °C, 10% (v/v) glycerol, 500 mM imidazole pH 8, 1 mM TCEP pH 8). The HisTrap column was then removed, and the Amylose column was washed with 5 CV lysis buffer. The Rpd3S complex was eluted with 5 CV Amylose Elution buffer (300 mM NaCl, 20 mM Na·HEPES pH 7.4 at 25 °C, 10% (v/v) glycerol, 30 mM imidazole pH 8, 116.9 mM maltose, 1 mM TCEP pH 8). Fractions containing intact Rpd3S complex were pooled,1.5 mg of TEV protease was added to remove the N-terminal His6-MBP tag, and immediately dialyzed into dialysis buffer (100 mM NaCl, 20 mM Na·HEPES pH 7.4 at 25 °C, 10% (v/v) glycerol, 30 mM imidazole pH 8, 1 mM TCEP pH 8) overnight at 4 °C. The next day, the

Rpd3S complex was applied to a 5 mL HiTrap Q (Cytiva) column equilibrated in dialysis buffer. The Rpd3S complex was eluted with a 20 CV gradient with high salt buffer (1 M NaCl, 20 mM Na·HEPES pH 7.4 at 25 °C, 10% (v/v) glycerol, 30 mM imidazole pH 8, 1 mM TCEP pH 8). Fractions containing intact complex were concentrated using an Amicon 100,000 MWCO centrifugal filter unit (Millipore). The concentrated sample was applied to a Superose 6 Increase 10/300 GL (Cytiva) equilibrated in gel filtration buffer (300 mM NaCl, 20 mM Na·HEPES pH 7.4 at 25 °C, 10% (v/v) glycerol, 1 mM TCEP pH 8). The elution was fractionated and analyzed by SDS-PAGE. Rpd3S-containing fractions were concentrated with an Amicon 100,000 MWCO centrifugal filter unit (Millipore), aliquots were frozen in liquid nitrogen, and stored at −80 °C.

### Expression and purification of acetylated lysine 16 histone H4
Expression of H4K16Ac was performed as previously described[21]. Briefly, H3-H4K16Amber plasmid and pACKRS plasmid were co-transformed into a C321Δ *E. coli* expression strain. Picked colonies were placed into LB and grown at 37 °C. Once the OD reached 0.8, 20 mM NAM and 10 mM acetyllysine were added to the flasks. After 30 min, 0.2 (w/v) % arabinose was added to the flasks. Cells were then grown at 37 °C for 16 h. Pellets were harvested, and then purification of the expressed histone was carried out as described[21].

### Expression and purification of histone H2A, H2B, and H3K36C
Histones H2A, H2B, and H3K36C were expressed and purified as described[41,42]. The H3K36C mutant was generated by 'Round-The-Horn site-directed mutagenesis.

### Preparation of H3K36Cme3
H3K36Cme3 was generated as described[33,43].

### Preparation of nucleosomal DNA
The following template with a TsprI site and modified Widom 601 were inserted into a pIDTSmart-Kan vector (IDT): 5′-ACG AAG CGT AGC ATC ACT GTC TTG TGT TTG GTG TGT CTG GGT GGT GGC CGT TTT CGT TGT TTT TTT CTG TCT CGT GCC AGG AGA CTA GGG AGT AAT CCC CTT GGC GGT TAA AAC GCG GGG GAC AGC GCG TAC GTG CGT TTA AGC GGT GCT AGA GCT GTC TAC GAC CAA TTG AGC GGC CTC GGC ACC GGG ATT CTG ATA TCG CGC GTG ATC TTA CGG CAT TAT ACG TA-3′ using the same approaches as previously described[44].

DNA was prepared using large-scale PCRs in the same way as previously described[44] with primers 5′-ACG AAG CGT AGC ATC ACT GTC TTG-3′ and 5′-TAC GTA TAA TGC CGT AAG ATC ACG CG-3′. The TspRI site was cleaved resulting in a 30 bp double-stranded DNA overhang on one side of the nucleosome and a 9-nt overhang single-stranded DNA followed by a 31 bp overhang on the other side of the nucleosome.

### Octamer formation and nucleosome reconstitution
Histone octamers were formed as described[41,42]. Nucleosomes were reconstituted using a salt-gradient dialysis. Nucleosomes were subsequently purified by native PAGE using a PrepCell (Bio-Rad) and concentrated using an Amicon 30,000 MWCO centrifugal filter unit (Millipore). Nucleosome concentration was quantified by absorbance at 280 nm. The molar extinction coefficient of the nucleosome was obtained by summing the molar extinction coefficients of the octamer and the DNA components at 280 nm.

### Complex formation for cryo-EM
100 µL of nucleosome (0.9 µM) and Rpd3S (1.8 µM) were mixed on ice for 1 h in S6 buffer (50 mM NaCl, 20 mM Na·HEPES pH 7.4 at 25 °C, 1 mM TCEP pH 8, 4 % (v/v) glycerol). Sample was then applied to a Superose 6 Increase 3.2/300 column (Cytiva) equilibrated in S6 buffer. Fractions were analyzed by SDS-PAGE and then briefly cross-

linked as previously described. Sample was dialyzed into dialysis buffer (50 mM NaCl, 20 mM Na·HEPES pH 7.4 at 25 °C, 1 mM TCEP pH 8) for 3 h.

Quantifoil R2/1 on 200 Mesh copper grids were glow discharged for 30 s at 15 mA using a Pelco Easiglow plasma discharge system. 4 μL of dialyzed sample was applied to grids for 8 s, blotted for 5 s with a blot force of 8, and vitrified by plunging into liquid ethane using a Vitrobot Mark IV (FEI) at 5 °C and 100 % humidity.

### Cryo-EM data collection
Cryo-EM data was collected on a ThermoFisher Titan Krios at 300 keV equipped with a Gatan K3 direct electron detector and a Gatan Bio-Quantum energy filter. Data collection was automated using EPU software. Data was collected at a pixel size of 0.822 Å with a defocus range of −0.5 to −2.25 μm. The data set yielded 32,564 micrographs with 50 movie frames at an exposure time of 1.92 s with an electron flux of 17.983 e$^-$ Å$^{-2}$ s$^{-1}$ for a total exposure of 51.1 e$^-$ Å$^{-2}$.

### Cryo-EM data processing
Initial data processing was performed in cryoSPARC (v4.1.0)[45]. Movie alingment was performed using patch motion correction followed by patch CTF estimation. Blob picker was then used and 8,797,734 particles were extracted at a box size of 450 pixels and binned by two. Particles were split into half (4,638,532 particles and 4,159,202 particles) and four rounds of heterogenous refinement were performed to enrich for Rpd3S bound nucleosome. Remaining Rpd3S-bound particles were combined, un-binned, and subjected to homogenous refinement. Global and local CTF refinements were performed, and after non-homogenous refinement of the remaining 481,141 particles, we obtained a 2.8 Å map (map A). Local masked refinements for the nucleosome (map B—2.7 Å) and Rpd3S (map C—2.9 Å) allowed us to visualize features at greater detail. To enrich for the Eaf3 CHD-H3K36Cme3 interactions, we performed masked classifications without image alignments in RELION[46] (v4.0) with masks encompassing Eaf3$_A$ CHD or Eaf3$_B$ CHD. After classifications, particles were re-imported to cryoSPARC and masked local refinement on the nucleosome provided clear density for Eaf3$_A$ CHD (map F—2.9 Å) and Eaf3$_B$ CHD (map E—3.0 Å). A similar approach was used to visualize the auxiliary Rco1$_B$−Eaf3$_B$ module: after classifications with a mask encompassing Rco1$_B$−Eaf3$_B$ in RELION, particles were re-imported to cryoSPARC and masked local refinement on Rpd3S provided clear density for Rco1$_B$−Eaf3$_B$ (map D—3.3 Å). After five rounds of heterogenous refinement in cryoSPARC enriching for particles containing Ume1, non-uniform refinement provided a map with clear density corresponding to Ume1 (map G—4.0 Å). A composite map of maps A, B, C, D, E, F and G was created using FrankenMap[47,48] (map H) and sharpened in cryoSPARC.

### Model building and refinement
AlphaFold generated models of Rpd3S components were manually rigid body docked into map H with the aid of *S. pombe* Sin3S (PDB 8I02) and locally adjusted in Coot[49]. The nucleosome (PDB 3LZ0) was rigid body docked into map H[22] and locally adjusted in Coot. DNA trajectories were de novo built in UCSF ChimeraX and locally adjusted in Coot. An arginine inserting into the nucleosome acidic patch could be modeled de novo with adjacent residues visible. Since the adjacent residues could not be identified, they have been labeled as unknown in the model. A lysine in the Rpd3 active site could be modeled de novo with three adjacent residues on both sides visible. Since the adjacent residues could not be identified, they have been labeled as unknown in the model. The resolution of Ume1 is not sufficient for confident assignment of residues, so it was not included in the PDB deposition. All models were subsequently real-space refined in PHENIX[50] with secondary restraints using map H.

### Rco1-H2A-H2B Alphafold multimer prediction
AlphaFold multimer predictions for the Rco1-H2A-H2B acidic patch interaction were generated with AlphaFold multimer predictions (v2.3.1). The predictions were seeded with sequences for *S. cerevisiae* H2A.1 (P04911), *S. cerevisiae* H2B (P02293), and *S. cerevisiae* Rco1 (residues 1-79, Q04779). The AlphaFold multimer prediction yielded two potential canonical arginine anchors (Rco1 residues Arg38 and Arg51).

### Figure generation
All figures were generated using Adobe Illustrator, GraphPad Prism, and UCSF ChimeraX v1.5[51]. Supplementary Fig. 8 used PDB https://doi.org/10.2210/pdb5NL0/pdb[52].

### Reporting summary
Further information on research design is available in the Nature Portfolio Reporting Summary linked to this article.

## Data availability
The cryo-EM reconstructions and final models were deposited with the Electron Microscopy Data Base (maps A-H; accession code EMD-41449) and with the Protein Data Bank (accession code 8TOF). Source data are provided with this paper.

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

## Acknowledgements

We thank Manuel Osorio Valeriano, Felix Steinruecke, and Sophie Roth for support with insect cell maintenance. We thank all members of the Farnung lab for discussions. We thank The Harvard Cryo-EM Center for Structural Biology at Harvard Medical School and the Cryo-EM Facility at MIT for support with data collection. S.M.V is supported by the NIH New Innovator Award (DP2-GM146254) and The Smith Family Awards Program for Excellence in Biomedical Research. L.F. is supported by The Smith Family Awards Program for Excellence in Biomedical Research, Rita Allen Foundation, the NIH New Innovator Award (DP2-ES036404), and the Damon Runyon Rachleff Innovation Award in Cancer Research.

## Author contributions

L.F. cloned the Rpd3S complex and performed bacmid and virus production. J.W.M. expressed and purified protein components and conducted all biochemical experiments. J.W.M. prepared the complexes for cryo-EM. J.W.M collected an initial dataset and S.M.V. collected the final cryo-EM dataset. J.W.M. processed all cryo-EM data with input from L.F. L.F. designed research. S.M.V. and L.F. supervised research. J.W.M., S.M.V. and L.F. wrote the manuscript.

## Competing interests

The authors declare no competing interests.
