## [Peer Review File · Nature Communications]

Structure of the complete *Saccharomyces cerevisiae* Rpd3S-nucleosome complexREVIEWER COMMENTS

Reviewer #1 (Remarks to the Author):

Summary:

The manuscript titled "Structure of the complete *S. cerevisiae* Rpd3S nucleosome complex" by Markert et al. presents the structural characterization of the yeast class I HDAC-containing Rpd3S complex and its interactions with a substrate nucleosome using single-particle cryo-EM data. The Rpd3S complex plays a crucial role in chromatin regulation, specifically in deacetylating lysine residues of histones H3 and H4, particularly in transcribed gene regions, thereby preventing spurious transcription events. HDAC enzymes operate within multi-subunit protein complexes that determine their location and activity regulation. Notably, unlike other HDAC complexes, Rpd3S/L complexes, in which the HDAC subunit interacts intimately with the scaffolding protein Sin3, do not require inositol phosphates for HDAC activity stimulation.

The study provides novel insights into the interactions within the Rpd3S complex and its interactions with nucleosomes. It also rationalizes previous biochemical findings using the presented structural data. The structural information is consistent with Rpd3S's broad substrate specificity regarding the lysine residues targeted by HDAC activity.

The manuscript is well-written, with clear and comprehensible figures. However, some figures/panels should be moved to the supplementary materials (see below). The text is logically structured and easily understandable. Notably, the authors elaborate well on the role of Rco1A in stabilizing the complex structure and the details of Sin3-Rpd3 interactions, supported by both data and existing literature.

In summary, this study is a valuable contribution to the field of chromatin regulation, suitable for Nature Communications' general readership. The findings have significant implications and are expected to attract substantial attention. However, certain limitations in the experimental data require addressing, with suggestions to better align the text and figures with the data quality and certainty levels, potentially making it a candidate for publication in Nature Communications.

Overall Quality of EM Densities:

The quality of the experimental EM densities, both globally and locally, introduces uncertainty into the authors' conclusions, which should be explicitly acknowledged in the manuscript.

The model constructed for the Rpd3S/nucleosome complex generally aligns well with the provided EM densities. However, it is essential to clarify where and to what extent the atomic model is supported by experimental data, especially for critical portions of the structure. While this information is provided in extended data table 3, it should also be included in the main text as remarks for clarity.

For example, the auxiliary module composed of Rco1B/Eaf3B is poorly resolved, and it is crucial to state that this portion of the model is based on alphafold predictions, confirmed by experimental data but lacking high-resolution details. In particular, in the absence of additional supporting data using biochemical / other structural biology methods, the relatively low local resolution introduced a degree of uncertainty. Given the power and reliability of alphafold prediction, it is very likely that the model is correct, but an open statement regarding the relationship to experimental data has to be provided. Highlighting the potential uncertainty introduced by the low local resolution is essential.

To facilitate analysis by the structural biology community, the authors should specify which maps from Suppl. Fig. 2 were used for model building/refinement for each model.

Ume1 Binding:

The identification of EM density corresponding to Ume1 adds valuable information to previous work on the Rpd3S complex. However, the limitations regarding Ume1's detection and its interface with Sin3/Rco1/Eaf3 should be acknowledged.

The limited data subset (~10%) and low local resolution (~10-15 Å) of the Ume1 density require careful consideration. E.g., it is not possible to reliably orient Ume1 with respect to the remainder of the complex. Therefore, the identification of Ume1 and its interface with Sin3/Rco1/Eaf3 would require further work, should it remain in the manuscript. For example, negative stain EM data of an Rpd3S complex that lacks Ume1 could provide a comparison with the complete Rpd3S complex, supporting the position and identity of Ume1 binding. Another possibility would be cross-linking mass spectrometry to provide further evidence that may support Ume1 binding to this side of the complex.

In the EM density and model, Ume1 is positioned some distance away from the interfaces proposed by the authors, except for the Eaf3B chromodomain. For example, it is not clear that the Sin3 HCR indeed makes contact with Ume1. Can any of the interfaces be predicted e.g. by alphafold? Another possibility would be to mutate the proposed interfaces and show biochemically, that Ume1 association with the complex is lost.

Moreover, parts of the Rpd3S subunits in this region are poorly resolved / cannot be modeled. Is it possible that regions of the respective proteins that cannot be detected in this map contribute to Ume1

binding? Would a flexible interface be a possible explanation for the poor local resolution and difficulties to unambiguously identify the position of Ume1 within the complex? The authors should implement these or similar considerations.

According to these limitations, the authors should tone down some of the statements in the manuscript. Statements like 'novel interface' or 'novel interaction' should be moderated due to the limited data.

Regarding nucleosome-bound Rpd3S complex stoichiometry, the potential presence of a second Ume1 subunit should be explored further. Negative-stain EM could offer additional evidence, and whether or not there would be a possibility of accommodating a second Ume1 through the same interface should be discussed.

Overall, I would suggest that the main figure 4 should be moved to the supplement. Unless further evidence is provided, Ume1 may also be omitted from the pdb file, and its putative position could rather be indicated in a figure rather than in the pdb.

Rpd3S Nucleosome Interactions:

The study effectively illustrates multivalent interactions with nucleosomes, providing insight into chromatin regulator-substrate nucleosome anchoring. However, certain binding sites lack strong data support, requiring adjustments in the manuscript.

The Rco1 – H3 loop L1 interactions are not well enough supported by the data. This is reflected by the statement in ll. 186-187 'residues...are near...'. However, this H3 histone fold interaction is also mentioned in ll. 47-48 and l. 178 as an important feature of Rco1 nucleosome interactions. Lacking further confirmation and clear EM density supporting this information, this interaction cannot be clearly concluded.

Rco1A R384 binding to the nucleosomal DNA is possible according to the model, but the EM density does not show or clearly support this interaction. A careful statement could be made, but Fig. 5d should be removed.

Regarding the interactions of the Eaf3 chromodomains with the nucleosomal DNA, the interaction is very clear for K85, and possible for K26. However, the C-terminal parts of the Eaf3 helices contacting the extranucleosomal DNA are poorly resolved due to the flexibility of the DNA in this region. Therefore, it is not possible to specify that, in this part of Eaf3, it is indeed these two residues that are important for Eaf3 binding to the extranucleosomal DNA. The text and figures are misleading since they suggest that this interaction was clearly shown and supported by the experimental data, which is not the case. The

authors should therefore choose more careful wording and representation of this interaction, since the modeling suggests such interaction, which is not entirely supported by the underlying data.

The acidic patch interactions are clearly supported by the density and represented with appropriate caution in the text. Since this is one of the interactions that was not reported by all of the other publications/preprints on Rpd3S nucleosome binding – what could be potential implications of acidic patch binding by the N-terminus of Rco1? Are any mutations known of the respective arginines that affect the function of Rpd3S? How is this region conserved. Additional experiments clarifying the impact of this interaction on nucleosome binding and HDAC activity would add value to the study.

In conclusion, this manuscript offers valuable insights into the Rpd3S nucleosome complex but must address limitations and uncertainties in the data presentation and discussion. Adjustments should be made to the text and figures to align with the data quality and certainty levels, and potential differences with other published structures, in particular that of Guan et al. (doi:10.1038/s41586-023-06349-1) that has passed peer-review previously, should be thoroughly explored and discussed.

Minor points / typos

- II. 112-113 – it should be electrostatic interactions, not hydrogen bonds
- I. 137 Eaf3B chromodomain, not Rco1B chromodomain
- I. 165 arginine 384, not lysine

Reviewer #2 (Remarks to the Author):

The Rpd3S complex, a histone deacetylase complex, can recognize H3K36me3 in transcribed regions and maintain chromatin integrity following Pol II elongation. Despite thorough investigations into Rpd3S's role during transcription elongation, the precise mechanisms underpinning the assembly of the Rpd3S complex and its interaction with its nucleosomal substrate for broad substrate specificity remain largely elusive. Recent studies have reported the structures of Rpd3S and its homologs in *S.cerevisiae*, *S.pombe*, and *H.sapiens*, and provided valuable insights into the functions of histone deacetylases across diverse species.

In this manuscript, the authors reveal the cryo-EM structure of the fully assembled Rpd3S complex bound to a nucleosome. This structure provides an unambiguous depiction of the Rpd3S assembly's coordination, highlighting how multiple subunits interact with the nucleosome via diverse interfaces to support Rpd3S's broad substrate specificity. The targeted nucleosome sites include two histone H3 K36me3, nucleosomal DNA, linked DNA, and acidic patches within the histone globular domain. Notably, for the first time, the authors observe the assembly of Ume1 in the Rpd3S complex, which is a distinction among recent similar studies. Even though Ume1 is shared between Rpd3S and Rpd3L deacetylase

complexes, its function within these complexes has largely remained indeterminate. The structural data presented here, along with the observations in Rpd3L (NSMB 2023), should facilitate future functional investigations. In addition, the identified structure pinpointed the exact amino acid residues in contact with the two Sin3 Glu fingers, suggesting Sin3's regulatory role in regulating Rpd3 catalytic activity, which aligns with previous findings in yeast Rpd3S and mammalian Sin3B complex.

Overall, the data presented are of exceptional quality and convincingly corroborate the authors' primary conclusions. Its publication should attract a wide readership for further examination and potential drug discovery.

Specific Comments:

1. Two Eaf3 units appear to interact with the linker DNA at the entry and exit points, respectively. Figure 5A in this manuscript supports this points (one Eaf3 on the Sin3 side, and another on the opposite side). However, in Figure 5E and 5F, the authors mistakenly label both copies as contacting the exact same SHL+7.5. This should be corrected for accuracy.
2. In lines 160 and 162, the term "SI domain" is referenced. For the sake of consistency, it would be more suitable to utilize "SID" or the "SID domain".
3. In line 137, "Ume1 binds Sin3 and the Rco1B chromodomain," should be revised to "Ume1 binds Sin3 and the Eaf3B chromodomain.
4. In line 150, the authors imply that Eaf3B is essential for Ume1's incorporation into Rpd3S (Ref 26). Although Eaf3 might contribute to stabilizing Ume1 within the complex, a previous study has reported that Ume1/Rpd3/Sin3 can form a core complex without the presence of Eaf3 (PMID: 22177115).
5. In line 88, the study cited in Ref 16 appears to only show two copies of Rco1 in Rpd3S in vitro, but not in an in vivo context.

Response to reviewers

Reviewer #1 (Remarks to the Author):

Summary:

The manuscript titled "Structure of the complete *S. cerevisiae* Rpd3S nucleosome complex" by Markert et al. presents the structural characterization of the yeast class I HDAC-containing Rpd3S complex and its interactions with a substrate nucleosome using single-particle cryo-EM data. The Rpd3S complex plays a crucial role in chromatin regulation, specifically in deacetylating lysine residues of histones H3 and H4, particularly in transcribed gene regions, thereby preventing spurious transcription events. HDAC enzymes operate within multi-subunit protein complexes that determine their location and activity regulation. Notably, unlike other HDAC complexes, Rpd3S/L complexes, in which the HDAC subunit interacts intimately with the scaffolding protein Sin3, do not require inositol phosphates for HDAC activity stimulation.

The study provides novel insights into the interactions within the Rpd3S complex and its interactions with nucleosomes. It also rationalizes previous biochemical findings using the presented structural data. The structural information is consistent with Rpd3S's broad substrate specificity regarding the lysine residues targeted by HDAC activity.

The manuscript is well-written, with clear and comprehensible figures. However, some figures/panels should be moved to the supplementary materials (see below). The text is logically structured and easily understandable. Notably, the authors elaborate well on the role of Rco1A in stabilizing the complex structure and the details of Sin3-Rpd3 interactions, supported by both data and existing literature.

In summary, this study is a valuable contribution to the field of chromatin regulation, suitable for Nature Communications' general readership. The findings have significant implications and are expected to attract substantial attention. However, certain limitations in the experimental data require addressing, with suggestions to better align the text and figures with the data quality and certainty levels, potentially making it a candidate for publication in Nature Communications.

We thank the reviewer for their careful consideration of our manuscript. We have addressed the reviewer's comments with point-by-point answers (see below):

Overall Quality of EM Densities:

1) The quality of the experimental EM densities, both globally and locally, introduces uncertainty into the authors' conclusions, which should be explicitly acknowledged in the manuscript.

The model constructed for the Rpd3S/nucleosome complex generally aligns well with the provided EM densities. However, it is essential to clarify where and to what extent the atomic model is supported by experimental data, especially for critical portions of the structure. While this information is provided in extended data table 3, it should also be included in the main text as remarks for clarity.

For example, the auxiliary module composed of Rco1B/Eaf3B is poorly resolved, and it is crucial to state that this portion of the model is based on alphafold predictions, confirmed by experimental data but lacking high-resolution details. In particular, in the absence of additional supporting data using biochemical / other structural biology methods, the relatively low local resolution introduced a degree of uncertainty. Given the power and reliability of alphafold prediction, it is very likely that the model is correct, but an open statement regarding the relationship to experimental data has to be provided. Highlighting the potential uncertainty introduced by the low local resolution is essential.

We thank the reviewer for their feedback. We have clarified within the text the confidence in our model building based on the quality of our cryo-EM maps. We now state that the Rpd3S core has been built with

high confidence in side-chain placement (lines 97-99), and the Eaf3B-Rco1B was built by rigid-body docking AlphaFold2 models into secondary structure features (lines 135-137). The reviewer is correct that the Ume1 density is of insufficient resolution to position Ume1 confidently. However, a global placement relative to other Rpd3S subunits is possible. We have clarified this in the manuscript (lines 145-147).

2) To facilitate analysis by the structural biology community, the authors should specify which maps from Suppl. Fig. 2 were used for model building/refinement for each model.

We thank the reviewer for giving us the opportunity to further clarify how we build and refined the model. We used map H (which is a Frankenmap of maps A, B, C, D, E, F, and G) for model building. This is now clearly pointed out in Supplemental Figure 2 and more thoroughly described in the methods section.

3) Ume1 Binding:

The identification of EM density corresponding to Ume1 adds valuable information to previous work on the Rpd3S complex. However, the limitations regarding Ume1's detection and its interface with Sin3/Rco1/Eaf3 should be acknowledged.

The limited data subset (~10%) and low local resolution (~10-15 Å) of the Ume1 density require careful consideration. E.g., it is not possible to reliably orient Ume1 with respect to the remainder of the complex. Therefore, the identification of Ume1 and its interface with Sin3/Rco1/Eaf3 would require further work, should it remain in the manuscript. For example, negative stain EM data of an Rpd3S complex that lacks Ume1 could provide a comparison with the complete Rpd3S complex, supporting the position and identity of Ume1 binding. Another possibility would be cross-linking mass spectrometry to provide further evidence that may support Ume1 binding to this side of the complex.

We appreciate the reviewer's feedback. Parts of this feedback has been addressed in reviewer comment #1. Furthermore, a study by Wang et. al. (PMID: PMC10115800) of the Rpd3S S. pombe homolog Sin3S in the apo-state visualized the corresponding Ume1 homolog at high resolution that allowed for accurate placement. This Sin3S model (PDB: 8I02) was used for initial docking of Ume1 into our Ume1 density (map G). The positioning of the Sin3S Ume1 homolog corresponds well with our observed density. After superposition, Ume1 only required subtle movement to fit into our density. We have generated an additional figure for the reviewer to visualize this (see below). Accordingly, we have clarified this point in the results section of the manuscript (lines 144-147). The observed interfaces of the Sin3S model have also been biochemically confirmed for the S. cerevisiae Rpd3S complex (cf. references in Ume1 paragraph in manuscript: Chen et al., 2012 (PMID: 22177115)).

We also note (based on a later point from this reviewer) that we have removed Ume1 from the final PDB to prevent over-interpretation of our structural model. We look forward to future research that will clarify the Ume1-Rpd3S complex interface further.

Response to reviewer Figure 1: Fitting *S. pombe* Sin3S into map H. *S. pombe* Sin3S (PDB 8I02) was rigid body docked into map H. The corresponding Prw1 domain (colored in orange) is near the extra density in map G (colored in green). Prw1 is the *S. pombe* homolog of Ume1. Ume1 was then rigid body docked into the corresponding density, using the original location of Prw1 as the starting point.

4) In the EM density and model, Ume1 is positioned some distance away from the interfaces proposed by the authors, except for the Eaf3B chromodomain. For example, it is not clear that the Sin3 HCR indeed makes contact with Ume1. Can any of the interfaces be predicted e.g. by alphafold? Another possibility would be to mutate the proposed interfaces and show biochemically, that Ume1 association with the complex is lost.

We appreciate the reviewer's interest in further understanding the Sin3-HCR interactions. As mentioned in the last comment, the S. pombe Sin3S complex was able to visualize the corresponding Ume1 homolog at higher resolution. In that structure, the authors can clearly visualize the Sin3 HCR domain engaging with Ume1 (see figure below). We were unable to visualize this interaction at sufficiently high resolution, therefore we are unable to place it in our final model, but the relative position of Ume1 to the Sin3 HCR domain is consistent in our data, suggesting that the Sin3 HCR domain interaction also exists in S. cerevisiae (as described in point 3 above). We have also added a statement of this in our results section in lines 147-150.

AlphaFold2 could not predict the interaction.

Response to reviewer Figure 2: *S. pombe* Pst2 HCR-Prw1 interactions. The *S. pombe* homologs for Sin3 (Pst2) and Ume1 (Prw1) bind each other via the HCR domain of Sin3 (Pst2) in PDB 8i02.

5) Moreover, parts of the Rpd3S subunits in this region are poorly resolved / cannot be modeled. Is it possible that regions of the respective proteins that cannot be detected in this map contribute to Ume1 binding? Would a flexible interface be a possible explanation for the poor local resolution and difficulties to unambiguously identify the position of Ume1 within the complex? The authors should implement these or similar considerations.

We commend the reviewer for this insightful prediction that regions not detected in our map could contribute to Ume1 binding. Indeed, parts of the Sin3 HCR domain (which we cannot visualize) are shown to interact with Ume1 in the corresponding S. pombe Sin3S PDB 8i02 (as discussed above). There is a 30 amino acid flexible linker between the last residue of the Sin3 HCR domain that we observe and the remaining HCR that is visualized in the S. pombe Sin3S (PDB 8i02) and interacts with Ume1. We have now added a statement describing exactly this in lines 147-150 of the manuscript.

6) According to these limitations, the authors should tone down some of the statements in the manuscript. Statements like 'novel interface' or 'novel interaction' should be moderated due to the limited data.

We agree with the reviewer, and we have now toned down the interpretations of Ume1 within the manuscript. Cf. lines 141-159.

7) Regarding nucleosome-bound Rpd3S complex stoichiometry, the potential presence of a second Ume1 subunit should be explored further. Negative-stain EM could offer additional evidence, and whether or not there would be a possibility of accommodating a second Ume1 through the same interface should be discussed.

We appreciate the reviewer's interest in further characterizing the stoichiometry of the Rpd3S complex. As biochemical data suggests, a second Ume1 strictly requires a second Sin3 subunit to have the same interface. As there is no evidence that a second Sin3 associates with the Rpd3S complex (indeed, a single Sin3 subunit is a hallmark of the Rpd3S complex), it is unlikely that a second Ume1 could associate with Rpd3S. We have added a sentence in the discussion that says future work will better characterize the stoichiometry (lines 242-245). Of course, action of Rpd3S complexes on a dinucleosome template could lead to the association of multiple copies of the Rpd3S complex.

8) Overall, I would suggest that the main figure 4 should be moved to the supplement. Unless further evidence is provided, Ume1 may also be omitted from the pdb file, and its putative position could rather be indicated in a figure rather than in the pdb.

We agree with the reviewer, we have moved Figure 4 to Extended Data Figure 7 and we have removed the Ume1 subunit from the PDB. Figures have been renumbered accordingly.

Rpd3S Nucleosome Interactions:

The study effectively illustrates multivalent interactions with nucleosomes, providing insight into chromatin regulator-substrate nucleosome anchoring. However, certain binding sites lack strong data support, requiring adjustments in the manuscript.

9) The Rco1 – H3 loop L1 interactions are not well enough supported by the data. This is reflected by the statement in ll. 186-187 ‘residues...are near...’. However, this H3 histone fold interaction is also mentioned in ll. 47-48 and l. 178 as an important feature of Rco1 nucleosome interactions. Lacking further confirmation and clear EM density supporting this information, this interaction cannot be clearly concluded.

As suggested by the reviewer, we have removed the relevant sections.

10) Rco1A R384 binding to the nucleosomal DNA is possible according to the model, but the EM density does not show or clearly support this interaction. A careful statement could be made, but Fig. 5d should be removed.

As suggested by the reviewer, we have removed the relevant sections, we have removed Figure 5d. Additionally, we have changed our phrasing in the manuscript to indicate Rco1A R384 is positioned near the DNA instead of directly indicating an interaction occurs (lines 172-174).

11) Regarding the interactions of the Eaf3 chromodomains with the nucleosomal DNA, the interaction is very clear for K85, and possible for K26. However, the C-terminal parts of the Eaf3 helices contacting the extranucleosomal DNA are poorly resolved due to the flexibility of the DNA in this region. Therefore, it is not possible to specify that, in this part of Eaf3, it is indeed these two residues that are important for Eaf3 binding to the extranucleosomal DNA. The text and figures are misleading since they suggest that this interaction was clearly shown and supported by the experimental data, which is not the case. The authors should therefore choose more careful wording and representation of this interaction, since the modeling suggests such interaction, which is not entirely supported by the underlying data.

As suggested by the reviewer, we have changed the wording for Eaf3 lysine residues 129 and 130 to better reflect the uncertainty of the data (lines 179-185). We have left the corresponding figure intact to give the community an opportunity to better characterize these suggested interactions in the future as the interaction with extranucleosomal DNA will present an exciting avenue to further characterize the action of Rpd3S in higher-order chromatin contexts.

12) The acidic patch interactions are clearly supported by the density and represented with appropriate caution in the text. Since this is one of the interactions that was not reported by all of the other publications/preprints on Rpd3S nucleosome binding – what could be potential implications of acidic patch binding by the N-terminus of Rco1? Are any mutations known of the respective arginines that affect the function of Rpd3S? How is this region conserved. Additional experiments clarifying the impact of this interaction on nucleosome binding and HDAC activity would add value to the study.

We thank the reviewer for the opportunity to discuss the Rpd3S-acidic patch interaction in further detail. To the best of our knowledge, no one has mutated these residues so far. Additionally, sequence alignments with the known *H. sapiens* homolog (PHF12) and *S. pombe* (*Pst2*) reveal little conservation compared to *Rco1* in the N-terminal region, and no conservation in the AlphaFold2 identified acidic patch binding residues (see figure below). However, recently we (and others) described how the human histone deacetylase Sirtuin 6 (*Sirt6*) was regulated by the acidic patch. In the case of *Sirt6*, we revealed that the acidic patch both promoted *Sirt6* recruitment to the nucleosome and deacetylase activity. We have now added a statement where we suggest a similar function for Rpd3S-acidic patch interactions (lines 195-198).

Response to reviewer Figure 3: Sequence comparison of *S. cerevisiae* Rco1 N-terminus with *H. sapiens* PHF12 and *S. pombe* Pst2. Comparison of *S. cerevisiae* Rco1 to *H. sapiens* PHF12 and *S. pombe* Pst2 reveals little conservation in the N-terminal region of Rco1, and no conservation in the AlphaFold2 identified acidic patch binding arginine residues (residues 38 and 51).

13) In conclusion, this manuscript offers valuable insights into the Rpd3S nucleosome complex but must address limitations and uncertainties in the data presentation and discussion. Adjustments should be made to the text and figures to align with the data quality and certainty levels, and potential differences with other published structures, in particular that of Guan et al. (doi:10.1038/s41586-023-06349-1) that has passed peer-review previously, should be thoroughly explored and discussed.

We thank the reviewer for these suggestions. We have modified our manuscript accordingly and have now added a statement in the discussion describing the key similarities and differences between our study and the already peer-reviewed study by Guan et. al. (lines 269-271).

Minor points / typos

- II. 112-113 – it should be electrostatic interactions, not hydrogen bonds

Corrected.

- I. 137 Eaf3B chromodomain, not Rco1B chromodomain

Corrected.

- I. 165 arginine 384, not lysine

Corrected.

Reviewer #2 (Remarks to the Author):

The Rpd3S complex, a histone deacetylase complex, can recognize H3K36me3 in transcribed regions and maintain chromatin integrity following Pol II elongation. Despite thorough investigations into Rpd3S's role during transcription elongation, the precise mechanisms underpinning the assembly of the Rpd3S

complex and its interaction with its nucleosomal substrate for broad substrate specificity remain largely elusive. Recent studies have reported the structures of Rpd3S and its homologs in *S.cerevisiae*, *S.pombe*, and *H.sapiens*, and provided valuable insights into the functions of histone deacetylases across diverse species.

In this manuscript, the authors reveal the cryo-EM structure of the fully assembled Rpd3S complex bound to a nucleosome. This structure provides an unambiguous depiction of the Rpd3S assembly's coordination, highlighting how multiple subunits interact with the nucleosome via diverse interfaces to support Rpd3S's broad substrate specificity. The targeted nucleosome sites include two histone H3 K36me3, nucleosomal DNA, linked DNA, and acidic patches within the histone globular domain. Notably, for the first time, the authors observe the assembly of Ume1 in the Rpd3S complex, which is a distinction among recent similar studies. Even though Ume1 is shared between Rpd3S and Rpd3L deacetylase complexes, its function within these complexes has largely remained indeterminate. The structural data presented here, along with the observations in Rpd3L (NSMB 2023), should facilitate future functional investigations. In addition, the identified structure pinpointed the exact amino acid residues in contact with the two Sin3 Glu fingers, suggesting Sin3's regulatory role in regulating Rpd3 catalytic activity, which aligns with previous findings in yeast Rpd3S and mammalian Sin3B complex. Overall, the data presented are of exceptional quality and convincingly corroborate the authors' primary conclusions. Its publication should attract a wide readership for further examination and potential drug discovery.

We thank the reviewer for their positive feedback. We have addressed all comments. See below for our point-by-point responses.

Specific Comments:

1. Two Eaf3 units appear to interact with the linker DNA at the entry and exit points, respectively. Figure 5A in this manuscript supports this points (one Eaf3 on the Sin3 side, and another on the opposite side). However, in Figure 5E and 5F, the authors mistakenly label both copies as contacting the exact same SHL+7.5. This should be corrected for accuracy.

We appreciate the reviewer's observation that Eaf3 is not labeled with +/- SHL values in figure 5e and 5f (now: Figure 4d, e). We intentionally label these as SHL 7.5 instead of labeling them as SHL -7.5 and SHL +7.5. We do not intend to suggest a directional binding of Rpd3S to the nucleosome. However, we understand that this labeling can be interpreted as SHL +7.5, so we have changed our figures 5e and 5f (now: Figure 4d, e) to the label SHL ±7.5.

2. In lines 160 and 162, the term "SI domain" is referenced. For the sake of consistency, it would be more suitable to utilize "SID" or the "SID domain".

Done as suggested.

3. In line 137, "Ume1 binds Sin3 and the Rco1B chromodomain," should be revised to "Ume1 binds Sin3 and the Eaf3B chromodomain.

Done as suggested.

4. In line 150, the authors imply that Eaf3B is essential for Ume1's incorporation into Rpd3S (Ref 26). Although Eaf3 might contribute to stabilizing Ume1 within the complex, a previous study has reported that Ume1/Rpd3/Sin3 can form a core complex without the presence of Eaf3 (PMID: 22177115).

We thank the reviewer for this important point. We have edited the manuscript accordingly: “Our data explain previous genetic and biochemical observations where it was observed that the Sin3 PAH3 domain was essential for Ume1 association with Rpd3S. The PAH3 domain coordinates Rco1 and Eaf3B of the auxiliary module, suggesting that the Ume1 interface we observe with Eaf3B could contribute for Ume1 incorporation into Rpd3S in the context of the nucleosome but is not necessarily required. Ume1, Rpd3, and Sin3 can form a complex in the absence of Eaf3.”

5. In line 88, the study cited in Ref 16 appears to only show two copies of Rco1 in Rpd3S in vitro, but not in an in vivo context.

Corrected.

REVIEWERS' COMMENTS

Reviewer #1 (Remarks to the Author):

The authors have responded to all concerns that were brought up and have modified the figures and text in ways that now better reflect the uncertainties that are associated with locally poor resolution and lack of supporting experimental evidence. I would therefore recommend the revised version for publication, although the lack of experimental data supporting some interpretations and functional implications (see first round of review) is limiting the strength of the work to some degree.

minor comments

I. 25 5-subunit should be five-subunit

I. 245 higher-ordered should be higher-order

Response to reviewers round 2

Reviewer #1 (Remarks to the Author):

The authors have responded to all concerns that were brought up and have modified the figures and text in ways that now better reflect the uncertainties that are associated with locally poor resolution and lack of supporting experimental evidence. I would therefore recommend the revised version for publication, although the lack of experimental data supporting some interpretations and functional implications (see first round of review) is limiting the strength of the work to some degree.

We thank the reviewer.

minor comments

- l. 25 5-subunit should be five-subunit
- l. 245 higher-ordered should be higher-order

Corrected.